# Key Advances in Solution Blow Spinning of Polylactic-Acid-Based Materials: A Prospective Study on Uses and Future Applications

**DOI:** 10.3390/polym16213044

**Published:** 2024-10-29

**Authors:** Nataša Nikolić, Dania Olmos, Javier González-Benito

**Affiliations:** 1Department of Materials Science and Engineering and Chemical Engineering, Universidad Carlos III de Madrid, 28911 Leganés, Spain; nnikolic@ing.uc3m.es (N.N.); dolmos@ing.uc3m.es (D.O.); 2Instituto Tecnológico de Química y Materiales “Álvaro Alonso Barba”, Universidad Carlos III de Madrid, 28911 Leganés, Spain

**Keywords:** polylactic acid, nanofibers, solution blow spinning, electrospinning

## Abstract

Solution blow spinning (SBS) is a versatile and cost-effective technique for producing nanofibrous materials. It is based on the principles of other spinning methods as electrospinning (ES), which creates very thin and fine fibers with controlled morphologies. Polylactic acid (PLA), a biodegradable and biocompatible polymer derived from renewable resources, is widely used in biomedical fields, environmental protection, and packaging. This review provides a theoretical background for PLA, focusing on its properties that are associated with structural characteristics, such as crystallinity and thermal behavior. It also discusses various methods for producing fibrous materials, with particular emphasis on ES and SBS and on describing in more detail the main properties of the SBS method, along with its processing conditions and potential applications. Additionally, this review examines the properties of nanofibrous materials, particularly PLA-based nanofibers, and the new applications for which it is thought that they may be more useful, such as drug delivery systems, wound healing, tissue engineering, and food packaging. Ultimately, this review highlights the potential of the SBS method and PLA-based nanofibers in various new applications and suggests future research directions to address existing challenges and further enhance the SBS method and the quality of fibrous materials.

## 1. Introduction

Polymeric materials are important substances with numerous applications, such as chemicals, electronics, optics, packaging, and biomedicine. Advances in science and technology and the continuous efforts of researchers in sourcing innovative and sustainable solutions have contributed to the development of polymer sensors, photovoltaic devices, and smart materials. As the modern world is increasingly dependent on consumer plastics, it is crucial to develop new materials that integrate with environmental concerns and support human health [1]. One of the leading research areas is finding a substitute for traditionally applicable plastics. The solution lies in developing and applying environmentally friendly materials that are compatible with living organisms, ensuring that they do not manifest any undesired effects on living beings and nature. Biopolymers have a high level of significance in medical applications, since they degrade into weak acids and other components that can be easily eliminated from or digested by the body [2]. They can belong to two groups of polymers, biodegradable and nonbiodegradable. Depending on the way they are produced, they can be bio-based or fossil-fuel-based. Bio-based polymers are yielded from plants, animals or microorganisms.

Among the different biopolymers, polylactic acid (PLA) is one of the most popular [3,4]. PLA polymers can be obtained from both bio-based and fuel-based resources, and they are mainly synthesized from renewable resources, including starch and sugarcane [5,6,7].

Porous materials manufactured from biopolymers, which are both biocompatible and biodegradable, are of significant interest for biomedical applications. The porosity in these materials expands their range of applications due to the increase of specific surface area, allowing for the transfer of different kinds of chemical agents. This characteristic is beneficial in applications such as drug release, exudate removal and nutrient transport. Among the polymers that can be processed as porous materials, PLA plays a crucial role in various medical applications because of its biocompatibility and biodegradability. Among these applications, tissue engineering, wound healing, cosmetics and drug delivery can be included [8,9]. Figure 1 shows a scheme in which most of the applications of porous PLA-based materials are collected.

Porous materials, more precisely those constituted by fibers that are based on biopolymers, are among the promising materials for biomedical applications due to their very high surface-to-volume ratio [11]. Their characteristics, such as high surface-to-volume ratio and small pore size, make them highly distinctive and suitable for a wide range of applications, including sensors, filtration, protective materials, biomedical devices, etc. [12]. Bio-based nanofibers’ lightweight nature and biocompatibility further enhance their influence as green materials in modern applications. Their tunable morphology enables structural and morphological modifications, including optimizing processing conditions, incorporating various nanoparticles, or combining more than one type of nanofibers, thereby expanding their potential use [13]. Nanofibers can undergo diverse modifications to enhance their optical, magnetic, thermal, and biological characteristics, making them suitable for applications in medicine and the electronics industry as smart materials [14,15].

Among the different techniques for producing nanofibrous materials, including drawing, self-assembly, and phase separation, electrospinning (ES) and solution blow spinning (SBS) are especially notable. They offer unique capabilities to produce continuous polymer fibers and the chance to control the diameter, orientation, porosity, and surface behavior of those materials [16].

Solution blow spinning is a relatively new method for producing fibrous materials. SBS generates fibrous materials through parallel streams of polymer solution and pressurized air, without the need for a high electric field, which is a driving force in the ES method. This absence of a high electric field, together with the high production rate and in situ applications, makes SBS very attractive and expands the range of applications to produce nanofibers [17,18,19].

In Figure 2, a bar chart is presented, which shows the increasing number of research papers published annually on “Solution Blow Spinning” from 2009 to 2024, thus indicating a growing interest in this innovative topic. Research publications have sharply increased since 2014. This upward trend emphasizes the importance of ongoing research in this area. Solution blow spinning has emerged as a processing technique with significant potential, and efforts should focus on exploring its various uses and applications while addressing current limitations to unlock new opportunities across laboratories and industries.

This review provides a detailed exploration of the solution blow spinning (SBS) technique for producing nanofibrous materials based on polylactic acid (PLA), highlighting their properties and potential applications. It aims to elucidate the intricacies of the SBS method, the characteristics of materials composed of extremely fine fibers, and the various potential uses of these materials.

## 2. Polylactic Acid (PLA)

Polylactic acid (PLA) is a biodegradable and compostable polymer gaining prominence across various industries due to its environmentally friendly properties [10]. Due to its biodegradable nature, PLA has become an increasingly popular choice for companies looking to reduce their carbon footprint and decrease their use of non-renewable resources. PLA belongs to the family of aliphatic polyesters, which are derived from α-hydroxy acids. One of its key advantages is that it can be produced from renewable resources such as corn starch or sugarcane, offering a sustainable alternative to conventional petroleum-based plastics. As a result, PLA is widely used in applications ranging from packaging materials to medical devices. PLA has potential applications in the medical field as a biocompatible device. Its high-modulus and high-strength properties make it a desirable material for creating prosthetics, implants, and other medical devices. Since it is a biodegradable polymer, PLA eliminates the need for subsequent surgeries to remove implants, reducing both the patient’s recovery time and the medical waste generated [20].

PLA is synthesized from lactic acid (2-hydroxy propionic acid), which exists in two enantiomeric forms: L-lactic acid and D-lactic acid. These two forms provide flexibility in tailoring the polymer’s properties for specific uses. The proportion of L- and D-lactic acid in PLA can be adjusted during synthesis to modify the polymer’s physical and mechanical characteristics to meet specific performance requirements [21]. The fundamental component for producing PLA, the building block, lactic acid, can be obtained naturally, or more precisely, by the fermentation of carbohydrates, or chemically, by applying synthesis. Lactic acid is unique because it contains both a hydroxyl (–OH) and a carboxyl (–COOH) group, allowing it to undergo self-esterification to form linear polyesters, like PLA. Unlike other hydroxyl acids, lactic acid does not form a lactone due to the proximity of these functional groups. Instead, it forms a dimer, as illustrated in Figure 3, and can be polymerized using ring-opening polymerization or direct condensation methods [22].

Lactic acid has an asymmetric carbon atom, allowing it to exist as two optically active isomers: L (+) and D (−). Both isomers can be synthesized by bacterial systems. The chemical structures of the L (+) and D (−) enantiomers are shown in Figure 4, illustrating their distinct stereochemical configurations [23].

Polylactic acid (PLA) has the capacity to control its stereochemical configuration of the polymer, thereby influencing the rate of crystallization, the level of crystallinity, the processing temperature, and the mechanical properties of the material. Microorganisms primarily produce L-lactic acid, while D-lactic acid is not widely available from natural sources. Consequently, commercially available PLA typically contains over 90% of the L isomer. In the ring-opening polymerization of lactide, the repeating units are either introduced as dimers during polymerization or added as lactic acid monomers through direct condensation polymerization [23,24]. Lactide can be obtained in various enantiomeric forms, including L, L or D, D enantiomers, commonly referred to as L- lactide and D- lactide, respectively. Additionally, lactide can be obtained as a meso-compound and as a racemic mixture, containing equal amounts of L and D enantiomers, and it is often denoted as DL-lactide, as shown in Figure 5. Some confusion exists in the literature regarding the terminology, as meso-lactides and rac-lactides are sometimes incorrectly used interchangeably. A meso-compound, also known as a meso-isomer, is a unique type of stereoisomer that is a non-optically active member of a set of stereoisomers. A meso-component contains at least two stereocenters that are optically active, yet the molecule is not chiral. On the other hand, a racemic mixture or racemate comprises equal amounts of enantiomers of an optically active molecule, with one enantiomer rotating the plane of polarized light to the right and the other enantiomer rotating it to the left [23,25].

Poly (L-lactic acid) (PLLA) and its isomer Poly (D-lactic acid) (PLDA) are crystalline polymers. In contrast, Poly (DL-lactide) (PDLLA), which consists of racemic lactide units, typically demonstrates an amorphous nature. The properties of PLA vary depending on the ratio of D/L units in the polymer chain. In the cases of PLLA and PDLA, the arrangement of L- and D- lactic acid units in the polymer chain impacts the crystallinity, with higher racemic content resulting in decreased crystallinity, and lower racemic content leading to increased crystallinity.

The crystallinity of PLA-based materials is an important characteristic to have under control, since it affects various properties of the polymer, such as its mechanical strength, thermal stability, and optical properties. Consequently, the relationship between racemic content and polymer properties can be useful in tailoring the performance of PLA-based materials for specific applications. When the concentration of L-lactic acid in PLA is more than 93%, the polymer is in a semi-crystalline state. However, when the concentration of L-lactic acid is between 50 and 93%, the polymer becomes amorphous. The presence of D- and meso-lactides in PLA results in polymer twisting, compared with the higher concentrations of poly(L-lactide), in which a very regular crystal structure is present [23,26]. PLA can crystallize in three different forms: α, β, and γ. Among these forms, the α-structure has the highest melting temperature (*T_m_*), at 185 °C, while the β structure has a slightly lower *T_m_* of 175 °C. This indicates that the α form is more stable compared to the β form [21,23]. Stereochemically pure Poly (lactide) (either L- or D-) has a practically possible melting point of approximately 180 °C. However, when meso-lactide or D-lactide is mixed from one into another into a PLA-based material, this lowers the melting point, as well as the rate and degree of crystallization in the final polymer. On the other hand, the glass transition temperature (*T_g_*) of the polymer remains largely unaffected. When the PLA has around 15% of meso-lactide in its composition, the resulting polymer can no longer crystallize and becomes amorphous in structure. Materials that require heat-resistant properties can be injection-molded using PLA containing less than 1% D isomer. Additionally, nucleating agents can be introduced to encourage the formation of crystalline structures under relatively short coupling cycles. Conversely, PLA resins with higher D isomer contents (4–8%) would be better suited for thermoformed, extruded, and blow-molded products, as they are easier to process when the crystallinity is reduced [21,27]. The tendency of the lactide monomer to undergo racemization to form meso-lactide can affect the optical purity and, thus, the properties of the obtained PLA material.

### 2.1. Thermal Behavior of PLA

Polylactic acid (PLA) exhibits distinct thermal properties that are closely linked to its molecular structure, stereochemistry, and processing conditions, with transition temperatures playing a crucial role in determining its mechanical and physical behavior. The transition temperatures of high-molecular-weight PLA significantly affect its physical characteristics, such as mechanical strength, rheological properties, density, and heat capacity. In the solid state, polylactic acid (PLA) can exist either as an amorphous or as a semicrystalline material, with its form being primarily determined by its stereochemistry and thermal history. For amorphous PLA, the glass transition temperature (*T_g_*) is a critical factor that defines the upper temperature limit for most commercial applications. The *T_g_* is the temperature at which amorphous PLA transitions from a rigid, glassy state to a softer, rubbery state. Above the *T_g_*, amorphous PLA becomes less stable, losing its mechanical integrity and dimensional stability.

On the other hand, when amorphous PLA is cooled below *T_g_*, it behaves as a glassy material, with the molecular chains becoming rigid and locked in place. In the case of PLA, *T_g_* is influenced by the molecular weight of the polymer and its optical purity, i.e., by the ratio of L and D isomers present in the polymer. The higher the optical purity of the PLA and the higher the molecular weight of the polymer, the higher the *T_g_* of the material will be (Figure 6) [21]. Increasing the proportion of the L isomer leads to an increase in the *T_g_*, at infinite molecular weight. The *T_g_* values for 100%, 80%, and 52% L isomer fractions are 60.2, 56.4, and 54.6 °C, respectively. Therefore, we can conclude that the *T_g_* values of a polymer decrease as the proportion of the L isomer decreases [21,24].

Tsuji and Ikada [24] reported a similar dependence, as shown in Table 1, which illustrates the effects of different copolymer proportions on glass transition and melting temperatures. These results indicate a clear dependence on the copolymer ratio, which is critical for designing materials with tailored thermal properties for specific applications [21].

Figure 7 shows the variation of the melting temperature *T_m_* with the proportion of meso-lactide. PLA that is stoichiometrically pure (either L or D) has a maximum *T_m_* of approximately 180 °C. Whereas the presence of meso-lactide in PLA can lower its melting temperature by up to 50 °C.

The dependence of *T_m_* on the proportion of meso-lactide can be approximated by the following equation [28]:*T_m_* ≈ 175 °C − 300*W_m_*(1)
where 175 °C is the melting temperature of PLA made from 100% pure L-lactide, *W_m_* is the meso-lactide fraction below 0.18 level, and 300 is a constant that depends on the specific characteristics of the PLA being used. Generally, the *T_m_* values for PLA fall within the range of 130–160 °C, which is lower than the melting temperature of pure L-lactide PLA.

### 2.2. Crystallinity of PLA

The crystallinity of polylactic acid (PLA) plays a critical role in determining its mechanical and thermal properties, with the desired level of crystallinity varying based on the specific requirements of its intended application. Whether a high or low degree of crystallinity is desirable in PLA depends on the specific requirements of its intended use. The formation of crystals in PLA may be beneficial or detrimental, depending on the desired properties and performance of the final product. In the case of injection-molded preforms, a high level of crystallinity is not desirable, as it could impede the subsequent blow molding process. The fast crystallization of the polymer would make it difficult to stretch the preforms during the blow molding process, leading to lower-quality final products. Conversely, in applications in which thermal strength is critical, such as injection-molded products, increased crystallinity will be desirable. Increased crystallinity leads to improved thermal properties, making the product more resistant to heat and deformation under load. When polymers are mechanically oriented, a process known as strain-induced crystallization can occur. This phenomenon is observed in a variety of manufacturing processes, such as the stretch blow molding of bottles, thermoforming of containers, production of oriented PLA films, and fiber spinning. The degree of crystallinity resulting from mechanical orientation is affected by the proportion of L and D isomers in the polymer. When subjected to increasing draw ratios, the level of crystallinity in amorphous PLA increases. Conversely, decreasing the stereoisomeric purity of the polymer leads to a reduction in crystallinity (Figure 8) [21]. Considering the spinning process, ES and SBS, all those issues should not be lost sight of as they may be key to understanding the final behavior and performance of the spun materials, since fibers arise from complex mechanisms of drawing.

## 3. Polymer Nanofibers

Polymer nanofibers have received significant attention due to their unique properties, including small size, high porosity, large surface area, small pore size, and superior mechanical characteristics. These materials are highly adaptable and can be easily combined with various fillers, such as drugs, antibiotics, and different types of nanoparticles. Nanofibers are characterized by having at least one dimension of 100 nm or less [30]. In recent decades, notable efforts have been made to create new materials that are biodegradable and biocompatible, able to regenerate damaged tissues or wounds, and repair their functions. Among various biomaterials, polymeric materials have emerged as potential candidates for a range of medical applications, including drug delivery systems, hyperthermia, wound healing, biosensors, and tissue engineering [31,32,33]. The large surface area and high porosity of nanofibrous materials are among their most significant advantages. Their extensive surface area makes them highly sensitive and interactive with various mediums, enabling effective surface functionalization [34,35]. As Horne et al. mentioned in their work, verifying the quality of fibrous materials requires considering numerous parameters. This includes the material selection, fiber diameter, fiber diameter distribution, orientation of fibers, and functionalization [36].

Continuous advancements in nanotechnology are revolutionizing materials and devices to be manufactured at the nanoscale [37]. This is due to their good compatibility, extracellular-matrix-mimicking organization, and possible surface functionalization [38,39,40]. PLA-based nanofibrous materials can be tailored for specific applications through various modifications. These include adjusting the processing conditions to achieve the desired fibrous morphology, incorporating different types of nanoparticles and biomolecules, and applying surface modifications. In terms of their morphological properties, the orientation of nanofibers plays a significant role in the final applications of those materials, and it is very important in biomedical applications for cell adhesion and wound healing. Several studies have already shown that the physical arrangement of nanofibers significantly affects the behavior of different types of cells, such as mesenchymal stem cells (MSCs) and skin fibroblasts [41,42]. In their work, Zhao et al. demonstrated that immobilizing hyaluronic acid (HA) onto electrospun PLA fibrous mats shows the ability to capture CD44-receptor-expressing cancer cells. Zhao et al. have conducted experiments on both random and aligned PLA mats and have achieved improved capture of cancer cells using aligned PLA fibers. Based on this work, it is important to highlight the potential use of these materials for capturing circulating tumor cells (CTCs) in the future, after adequate nanofiber design and modifications [43].

When describing the role of PLA morphologies in the field of drug delivery and wound healing, it is important to mention that their small pores and high specific area make them highly effective for these applications [30]. Manipulating nanofiber content and porosity is crucial for achieving effective drug concentrations at the site of injury [44]. Drug release from nanofibrous mats can be controlled—delayed, immediate, rapid, or modified—depending on the polymer composition and pore size [45]. Due to these properties, nanofibers hold significant potential in wound healing, providing a physical barrier, absorbing excess exudates, and facilitating gaseous exchange. When loaded with antibiotics, they also exhibit antibacterial properties, enhancing wound protection [30,46].

## 4. Processing of Nanofibrillar Polymeric Materials

Among the various nanofiber fabrication techniques, such as template synthesis, self-assembly, drawing, and phase separation, electrospinning and solution blow spinning have emerged as particularly significant. Their growing importance is attributed to their versatility in producing nanofibers from a broad spectrum of materials, including polymers, metals, composites, and ceramics.

**Templating** is a standard method used to control the hierarchical structure of nanostructured surfaces. This method involves synthesizing the desired material with controlled dimensions and morphologies using a pre-designed template. The template is typically a membrane or a structure with predefined pores or channels. The desired material is introduced into the pores or channels of the template through various techniques, such as electrodeposition, sol–gel process, melt infiltration, and extrusion. Once the material is in the pores, it is solidified, and the template is then removed to release the synthesized nanostructures [47,48,49]. Nanofibers with long fiber lengths cannot be obtained using this method; the fibers produced are only a few micrometers in length. The diameter of these fibers is determined by the pore size of the membrane used. One of the advantages of template synthesis is its ability to fabricate nanofibers with different diameters by utilizing different templates [50,51,52].

The **self-assembly method** is a bottom-up nanomaterial fabrication technique that leverages non-covalent interactions, such as hydrogen bonding, hydrophobic forces, and electrostatic interactions. These forces drive the spontaneous organization of molecules into defined patterns or structures. This approach can produce nanofibers with diameters ranging from a few nanometers to less than 100 nm and lengths extending to several micrometers. Intermolecular forces facilitate the aggregation of molecular units, with the shape of these units dictating the final macromolecular structure of the nanofiber. The key advantages of self-assembly include its ability to generate extremely small nanofibers and to create complex structures at the nanoscale. However, the method is inherently complex, characterized by low productivity, and restricted to specific types of molecules [50,53,54].

**Drawing** is another technique for producing nanofibers, characterized by its ability to create a single fiber using a sharp tip or micropipette. The sharp tip is employed to draw a droplet from a previously deposited polymer solution, forming a liquid fiber. As the solvent evaporates due to the high surface area, the liquid fiber solidifies [53]. In this process, a micromanipulator dips the micropipette into the polymer solution droplet, which is then gently pulled from the liquid and moved at a low speed (~10^−4^ m·s^−1^). As the micropipette touches the substrate, the nanofibers are deposited, as depicted in Figure 9 [51].

Continuous nanofibers in various arrangements can be produced using this method. The key advantage of the drawing method is precise control over the drawing speed and viscosity, allowing repeatability and control over the dimensions of produced fibers [56]. However, the drawing method is a discontinuous process limited to a laboratory scale, as nanofibers are formed one by one. Consequently, it has very low productivity and is restricted to viscoelastic materials capable of withstanding the stresses generated during pulling. Additionally, the method is limited to producing fibers with diameters larger than 100 nm, depending on the orifice size [52].

The **phase separation** method relies on the physical incompatibility of components to induce separation into distinct phases. This method comprises four basic steps. First, the dissolution of a polymer takes place in a solvent at room or elevated temperature. Second, gelation is induced, which is crucial for controlling morphology, particularly porosity; the duration of gelation depends on the polymer concentration and gelation temperature. Subsequently, solvent extraction from the gelation using water is necessary, followed by freezing and vacuum freeze-drying [52]. The porosity of the resulting nanofibers correlates with the polymer concentration; lower concentrations increase fiber porosity but compromise mechanical properties [51]. The initial step involves preparing a homogeneous polymer solution at room temperature, followed by gel formation at a specific gelation temperature, through which nanofibrous matrices form due to phase separation. Solvent extraction and drying then yield nanofibers, as depicted in Figure 10 [52]. This method offers advantages such as the minimal equipment required and adjustable mechanical properties through polymer concentration [57]. However, it cannot produce long continuous fibers, and it is limited to certain polymers capable of undergoing phase separation to form nanofibers, such as PLA and polyglycolide, owing to their gelation capabilities [50,52].

### 4.1. Electrospinning

Electrospinning (ES) is a simple and effective technique to produce submicron- and nanoscale polymer fibers [58]. The electrospinning device consists of a high-voltage power supply, a syringe pump with a metal needle, and a grounded collector, which can be either a flat plate or a rotary drum. During the electrospinning process, a polymer solution is dispensed from a syringe pump, forming a droplet at the tip of a needle. To induce the deformation of the droplet into a conical shape, an electric field is applied between the needle tip and a grounded collector. In this way, because of electrostatic forces, the surface of the droplet or the small volume of the polymer solution is deformed into a conical shape, known as a Taylor cone. Since there is a critical surface tension in the polymer solution, it is necessary for the applied electric field to overcome it. Once the surface tension of the polymer has been overcome, the charged stream of the polymer is ejected from the Taylor cone toward the collector. The stream of a polymer solution travels toward the collector as a spiral that expands in this direction, toward the collector as described in Figure 11 [59,60,61].

A polymer jet can be affected by various forces that can contribute to its deterioration. The electrostatic force, repulsion force, surface tension, gravity, and viscoelastic forces influence the charged jet Figure 12 [62]. The polymer jet undergoes a gradual reduction in thickness as the solvent evaporates. Once the solvent evaporates, continuous fibers are collected on a collector varying from micrometers to nanometers in diameter [60,61,63,64].

A variety of solvents, including dichloromethane, acetone, chloroform, and tetrahydrofuran, can be employed in the production of PLA nanofibers. However, to minimize toxicity, it is advisable to use a mixture of solvents when dissolving PLA [65]. Highly porous PLA nanofibers are preferred for their potential to enhance characteristics crucial in applications such as tissue engineering, drug delivery, and wound dressing. In their research, Huang and Thomas [66] conducted a study on the formation of porous electrospun fibers using a mixture of solvents and non-solvents with a difference in volatility and miscibility. The compatibility between a solvent system and a polymer can be clarified through the utilization of Hansen solubility parameters. Specifically, when the Hansen solubility parameters (HSP) of both the solvent and the polymer closely align, this indicates a higher degree of compatibility between them. The HSP for PLA is 21.2 MPa^1/2^ [67]. In the context of ES, another critical attribute of the polymer solution is its dielectric constant. The dielectric constant (ɛ) is one of the most significant parameters and characteristics of solvents, since it is related to the polarity of the solvent. Solvents exhibiting a higher dielectric constant, ɛ, and, at the same time, greater polarity possess an increased capacity for carrying charge, consequently amplifying jet stability (Table 2) [66]. Manufacturing continuous fibers from PLA solutions involves various influential factors. These include the polymer concentration, electrical conductivity of the PLA solution, viscosity, and surface tension of the solution, as well as the entanglement concentration (C_e_). The concentration of the polymer solution plays a pivotal role in the electrospinning process, particularly in achieving nanofiber formation with a minimum critical concentration (C_e_) [68,69]. Below this critical concentration, droplet formation occurs instead of fiber formation during the process.

Conversely, for polymer concentrations above C_e_, the diameter of the resulting nanofibers increases while the occurrence of bead formation decreases. As already mentioned, Ce depends on factors such as the molecular chain length of the polymer, the chemical nature of the polymer, and the solvent composition employed in the polymer solution [68,69,70,71,72,73,74,75]. Several studies have illustrated that incorporating inorganic salts into polymer solutions enhances the net charge density of the electrospinning jet, facilitating the production of thinner fibers without bead formation. Greater net charge density amplifies the electrostatic repulsion within the jet, thereby promoting greater plastic stretching during electrospinning and, consequently, yielding thinner fibers [69,71,76,77,78,79]. In many cases, obtaining very thin fibers with a diameter within the nanoscale is preferable. Casasola et al. [80] have demonstrated a correlation wherein the fiber diameter decreases as the boiling point of a second solvent in a binary solvent system increases (Figure 13). Solvents possessing a higher boiling point undergo the evaporation process more slowly on the way from the nozzle to the collector. The slower evaporation of the solvent causes a change in the viscoelastic properties of the jet by stretching the jet so that the resulting fibers are much thinner.

Casasola et al. [80] have demonstrated that employing a binary solvent mixture of acetone and dimethylformamide (Ac/DMF) results in PLA fibers with reduced diameters and a more uniform distribution (10% *w*/*v* PLA solution), as compared to alternative binary solvent systems. The optimal composition for achieving the smallest and most uniform fiber diameter (272 ± 58 nm) was found to be 60/40 Ac/DMF.

Electrospinning (ES) exhibits several limitations, the main ones being a low production rate and the use of a high electric field. According to the production rate, multi-needle spinnerets are already used. However, the problem of needle blockage remains. On the other hand, high electric field is a critical parameter due to its direct interactions with the solvent used to dissolve the polymer solution. During processing, there are significant interactions between the applied voltage (high electric field) and the solvent. Considering these interactions, it is crucial to select a solvent that can withstand a high electric field and thereby produce continuous fibers without breaking from the needle tip to the collector. Organic solvents possessing high polarity, elevated boiling points, and considerable dielectric constants demonstrate strong compatibility with high electric fields. These characteristics enable them to effectively convey a charged jet of a polymer solution, thereby facilitating the production of continuous fibers [81,82,83,84].

Bubble electrospinning is a variation of the traditional electrospinning process, designed as an alternative to producing nanofibers more efficiently [85,86]. While both methods rely on the application of a high-voltage electric field to create fibers from a polymer solution, bubble electrospinning introduces a unique mechanism to produce nanofibers [87]. The basic working principle of bubble electrospinning is the formation of polymer bubbles instead of the Taylor cone observed in electrospinning. The electrical force coming from a high-voltage power supply should be set so that it overcomes the surface tension, thus causing multiple jets to eject, leading to the formation of the fibers in the collector [86,87]. A key challenge in this method is controlling the number, size, geometry, and stability of the bubbles, which has led researchers to propose “critical bubble spinning”, aiming to utilize a single, independent, and stable bubble for improved fiber production [88]. Bubble electrospinning can accommodate a wider range of polymer solutions, making it a versatile method for nanofiber fabrication. Its ability to produce nanofibers at a larger scale makes it particularly attractive for the mass production of nanofibrillar materials for applications in fields such as biomedical engineering, filtration, and energy storage, in which high volumes of nanofibers are often required [86].

### 4.2. Solution Blow Spinning (SBS)

In 2009, Medeiros et al. [89] proposed the solution blow spinning method (SBS) as a favorable alternative to electrospinning. Solution blow spinning is closely similar to electrospinning (ES), yet it distinguishes itself by employing pressurized air instead of a high electric field for stretching a polymer solution. This difference opens a broader range of solvents whose use is important in processing nanofibers. The absence of a high electric field might be an advantage of the SBS method, as it does not require specific interactions between the solvents, and a high electric field is not required [90]. One more key advantage of SBS over ES is its ability to achieve a higher production rate of nanofibers and films while maintaining comparable quality to those obtained through ES [89,91]. Nanofibers manufactured by SBS usually exhibit less homogeneous morphology in terms of fiber orientation and fiber diameter distribution. Nevertheless, SBS represents a scalable method conducive to the expansion of industrial material production, enabling large-scale manufacturing of the material [92].

One very important parameter for controlling the morphology of materials is the linear speed of the rotational collector. In the study conducted by J. González-Benito et al. [90], the linear speed of the collector was selected as the variable for modifying the morphologies of fibrous materials. The following equation was used to calculate the linear speed:v (m · s^−1^) = w (rad · s^−1^) · r(m)(2)
where w represents the angular velocity in radians per second and r is the radius of the collector in meters, where 1 rpm corresponds to 2π/60 = 0.1047 rad·s^−1^. The rotational speed of the collector varied from 200 rpm to 1000 rpm in J. González-Benito et al.’s study. The polymer for producing the nanofibers was poly (ethylene oxide). The researchers demonstrated that as the linear speed of the collector increases, the fibers become thinner, less entangled, and more oriented. Moreover, they observed enhanced fiber homogeneity in terms of size and a reduced number of joins, pointing out that the porosity of the materials increases, which is another crucial characteristic.

Figure 14 illustrates the schematic representation of the SBS process, containing the essential elements of the SBS device, including the pressure regulator, flow meter, syringe, syringe injection pump, and collector. As previously mentioned, the SBS method facilitates the production of fibers through the parallel stream of a polymer solution and pressurized air. Within the SBS setup, the polymer solution is placed inside the inner chamber of a concentric nozzle, while pressurized air occupies the outer chamber. The pressurized air induces the stretching of the polymer solution toward a collector. During this process, solvent evaporation occurs along the path from the nozzle to the collector, depicted in Figure 14 as the “working distance”. Consequently, the polymer solution undergoes rapid transformation into fibers, which are subsequently collected on the collector [89,93,94].

In their study, Oliveira J.E et al. [95] investigated the impact of solution viscosity on fiber diameter and its distribution. Their findings revealed that an increase in solution viscosity resulted in thicker fibers with a wider diameter distribution. The researchers attributed this phenomenon to the heightened difficulty in stretching the fibers due to elevated viscosity, thereby leading to the production of thicker fibers with a broader diameter distribution.

In the case of SBS, air pressure acts as a driving force for stretching the polymer solution and is also responsible for the distribution of the fiber diameter [96]. Even though higher air turbulence due to increased airflow might be expected to lead to a larger distribution of fiber diameters, Oliveira J.E [95] et al. found that higher airflow was necessary for polymer solutions with higher viscosity. This is again related to the stretching of the polymer solution; in their work with a PLA polymer, they achieved the narrowest fiber diameter distribution with the solution with the lowest viscosity. Another critical parameter of fibrous materials is their preferential orientation, which has a high impact on the final properties of the material. In their study, J. Benito et al. [90] highlighted the importance of the collector’s rotational speed. They experimented with rotational speeds of 200 and 1000 rpm using both small and large collectors. By considering both the rotational and the linear speed of the collector, which depend on the collector’s size, they achieved a clear monomodal preferred orientation of fibers with the large collector at 1000 rpm. If we consider the method used for collecting the materials, a bimodal distribution is actually expected. The nozzle moves parallel to the long axis of the cylindrical collector, which rotates perpendicular to the nozzle’s movement. When the nozzle moves to the left, one preferred orientation of fiber angles becomes favorable, while a different preferred orientation is obtained when it moves to the right (Figure 15) [90]. As the speed of the cylindrical collector increases, the difference between the two preferential orientations should decrease, since the collector’s speed becomes much higher than the nozzle’s movement.

Another critical parameter that significantly affects the morphology of fibrous materials is polymer concentration. In the SBS method, similarly to ES, the entanglement concentration (C_e_) is crucial for avoiding bead formation in the final material morphology. It is often proposed that Ce ≈ 10c*, where c* is the overlap concentration of the polymer solution at which individual polymer chains in the solution start to overlap and interact with each other [97]. Below this concentration, polymer chains are isolated, and the solution behaves as a dilute solution. However, when the concentration reaches c*, polymer chains start to overlap, leading to entanglements between them. At and above c*, the entanglements between polymer chains provide stability to the polymer jet formed during SBS processing, helping to overcome the forces that would otherwise lead to the formation of beads instead of continuous fibers [98]. Therefore, the solvent used should be able to dissolve the polymer successfully up to at least the overlap concentration. To minimize bead formation, the polymer concentration in the solution should be greater than c*, reaching Ce or higher (Ce ≈ 10c*) [99].

Oliveira J.E et al. [100] also investigated the impact of the polymer concentration on the final material morphology, particularly focusing on the effect on the fiber diameter. The concentration of the PLA solution was varied from 4% to 8%, resulting in an increase in fiber diameter as the concentration of PLA increased. They also examined the processing conditions of the SBS technique, such as pressurized air and feeding rate. The PLA-based nanofibers exhibited higher fiber diameter and distribution with an increase in the feeding rate of SBS process. The optimal feeding rate is one that yields fibers with a narrower distribution and smaller diameter. A high feeding rate can cause the blocking of the nozzle, while a very low feeding rate leads to jet instability. However, the feeding rate is highly dependent on the viscosity, which is influenced by polymer molecular weight and polymer concentration.

Pressurized air is another parameter affecting fiber diameter and distribution. An increase in air pressure enhances solvent evaporation during processing. Oliveira J.E. et al. demonstrated that higher pressurized air decreases fiber diameter and distribution, while lower pressure increases both fiber diameter and distribution, and can also cause more frequent nozzle blockages.

In their study, M. Mobaraki et al. [101] explain the ease with which SBS can be employed for in situ applications. Utilizing just an airbrush and compressed gas, this method enables the direct deposition of nanofibers onto diverse surfaces. Such versatility makes it applicable to numerous medical applications [102]. By adjusting the polymer concentration, properties, and processing conditions, fibers can be spun to achieve a large surface area, making them suitable for various applications, such as filtration, membranes for biological and chemical sensors, drug delivery, and tissue engineering [89,103].

## 5. Applications of PLA Nanofibrous Materials

A wide range of PLA-based nanofibrous materials have been investigated for biomedical applications, including drug delivery systems, wound healing, tissue engineering, etc. This is due to their unique characteristics, including a large specific surface, high porosity with small size of pores, and similarity to the extracellular matrix [38]. PLA-based nanofibrous materials can be tailored for specific applications through various modifications. These include adjusting the processing conditions to achieve the desired fibrous morphology, incorporating different types of nanoparticles and biomolecules, and applying surface modifications.

**Wound healing**—The properties of PLA fibers, such as large specific area, high porosity, good ductility, and excellent biological characteristics, are crucial for cell proliferation and protecting against bacterial infection, as well as for promoting faster wound healing. Traditional dressings lack these properties, often resulting in poor cell respiration, wound infections, and slow healing rates. The use of PLA-based fibers in wound healing aims to successfully absorb wound exudates and sustain an optimal moist environment for healing [38,104]. Many researchers have explored the potential of PLA-based membranes as dressing materials, demonstrating their capability to be loaded with different drugs and achieve successful drug release [105]. Anti-inflammatory agents such as betamethasone, dexamethasone acetate, and curcumin have already been loaded into PLA membranes [105,106]. Doxycycline has demonstrated potential as an antibiotic within PLA nanofibers, exhibiting beneficial cytocompatibility with L929 mouse fibroblasts, making it highly suitable for wound dressing applications [107]. Ilomuanya et al. [108] were successful in preparing non-toxic composite fibers by incorporating Aspalathus linearis fermented extract and silver sulphadiazine to enhance the antibacterial activity of materials.

**Drug Delivery**—PLA is widely used in biomedical applications due to its excellent biodegradability and biocompatibility. Approved by the Food and Drug Administration (FDA), PLA serves as a biomaterial in various applications, including drug delivery systems, bone plates, abdominal meshes, and sutures [109]. Several mechanisms, such as coating, embedding, and encapsulating, can be used for drug loading and release from nanofibers to control drug release kinetics [110,111,112]. Regarding the mechanism for mixing drugs with polymers, several methods can be employed. If the drug and polymer are dissolved in the same solvent, the drug can be added directly to the polymer solution and fibers can be spun. However, if the drug is not dissolved in the same solvent as a polymer, a drug can be dissolved in a different solvent before being added to the polymer solution. Another method involves dissolving the drug in a solvent that is not miscible with the polymer’s solvent. In this case, both solutions are introduced in a separate syringe to be spun coaxially or spun from the same syringe as an emulsion [113,114,115]. By changing the type of polymer and modifying the fiber’s diameter, porosity, and overall morphology, the drug release kinetics can be adjusted. Additionally, manipulating various processing parameters during the production of fibrous materials can further influence the drug release kinetics [112,116]. Finally, several mechanisms have been explored for releasing a drug from the material, such as diffusion through the canals and pores of nanofibers, desorption from the material’s surface, and matrix degradation [117,118].

**Tissue Engineering**—The characteristics of nanofibrous PLA materials, such as their very small diameter, which matches that of extracellular matrix (ECM), and their large surface area, which is conducive to cell attachment and bioactive loading, make PLA-based materials highly attractive for tissue engineering. To enhance the strength of PLA-based materials, they can be combined with other bioactive polymers to meet specific property requirements and applications [119]. Currently, innovative approaches are increasingly being used to design materials that mimic the nature of human tissue. A huge number of tissues in our body exhibit piezoelectric effects. For instance, bone is a tissue that has a piezoelectric constant similar to that of quartz, a major component of bone’s organic ECM. PLA is a polymer that belongs to a family of piezoelectric materials and, in that sense, it is particularly interesting for supporting tissue engineering [120,121,122,123]. In recent years, there has been an increase in the design of fibrous materials for tissue engineering, specifically in musculoskeletal, neural, and cardiovascular tissue engineering. Cardiovascular tissue engineering has attracted significant attention due to the high prevalence of cardiovascular diseases, which are among the leading causes of death worldwide. This is why many researchers are focused on producing fibrous materials with very small diameters and aligned nanofibers, to improve cell proliferation compared to randomly oriented fibers [119,124,125,126].

**Food packaging**—In comparison to the petroleum-based polymers used for packaging, PLA reveals several advantages, such as good transparency, degradation in a biological environment, processability, and biocompatibility. PLA is accepted by the Food and Drug Administration (FDA) for its biocompatibility and capability to degrade into non-toxic components [127]. Furthermore, food packaging materials must possess essential qualities, including optimal mechanical properties, thermal stability, water and gas vapor barrier properties, etc. PLA-based materials designed for food packaging hold significant promise for practical implementation. These versatile materials are used in producing containers for a range of applications, including milk and beverage bottles and fresh and cooked food. Additionally, PLA-based straws offer an eco-friendly alternative, substituting traditional paper or plastic straws in beverage shops, thereby making a meaningful contribution to environmental conservation [65].

**Micro-electromechanical systems (MEMSs)**—These are systems that operate at the micro or submicrometric scale, capable of exhibiting mechanical or electrical responses when subjected to electrical or mechanical perturbations, respectively [128]. As previously mentioned, polylactic acid (PLA) can demonstrate piezoelectric properties, making it particularly advantageous for integration into more complex MEMS devices utilized as sensors or actuators. Despite their ubiquitous presence in daily life, recent years have seen extensive research focused on the application of MEMS in the biomedical field [129]. This is a domain where PLA could be particularly beneficial due to its biocompatibility.

## 6. Conclusions and Future Prospectives

The solution blow spinning (SBS) method shows us numerous advantages over other nanofiber production techniques. Its high production rate and in situ applications make it very popular in various applications. By varying processing parameters, SBS can produce materials with different morphological characteristics, allowing control over fiber size, porosity, and orientation. The main advantage of SBS over the rest of the other relevant methods is its lack of high electric field and high production rate. This makes SBS particularly valuable for potential applications in drug delivery and wound healing, in which materials can be directly deposited with various drugs and antibiotics onto targeted organs, wounds, or tissues.

PLA, as a biodegradable and biocompatible polymer, holds significant promise for future medical applications. Its properties, such as crystallinity and thermal behavior, can be precisely tailored based on the Land D isomers present in the structure, opening a wide range of potential applications. As a green material, PLA plays a crucial role in modern technologies aimed at reducing the carbon footprint of plastics. The versatility of PLA is especially prominent in medical applications such as tissue engineering, wound healing, and drug delivery systems, emphasizing its capacity to enhance healthcare solutions. In the contexts of wound healing and tissue engineering, PLA nanofibers can serve as scaffolds that promote cell growth and offer controlled degradation rates. Additionally, in drug delivery applications, PLA’s adjustable degradation and release profiles significantly improve therapeutic outcomes by facilitating precise control over drug release rates.

Additionally, coupled with the potential piezoelectric properties of PLA, these characteristics are highly desirable for numerous micro-electromechanical system (MEMS) applications, particularly where devices may interface with biological systems, such as in monitoring vital signs or wearable health devices. The ability to precisely control the spinning process enables the fabrication of microscale fibers with specific structures, thus tailoring properties that can be integrated into MEMS sensors or actuators. Piezoelectric nanofibers are currently extensively used as piezoelectric biosensors in ultrasensitive MEMS systems, highlighting the potential of the solution-blown spinning of polylactic acid-based materials to support the development of advanced MEMS devices with enhanced functionality and biocompatibility.

In addition to medical uses, PLA-based materials have substantial potential in food packaging. PLA’s biocompatibility, biodegradability, and ability to form effective barrier properties make it a strong candidate for sustainable packaging solutions. Its use in food packaging can extend shelf life, reduce plastic waste, and enhance food safety through the incorporation of antimicrobial or other functional agents. Furthermore, the controlled degradation of PLA ensures that these packaging materials contribute to environmental sustainability, aligning with global efforts to minimize plastic pollution.

To fully realize the potential of the SBS method, it is necessary to develop a new system to enhance the quality and functionality of nanofibrous materials. Finding the best conditions, depending on the polymer and solvent system, is essential to achieve desired material properties, including morphology, surface behavior, thermal behavior, and structure. The challenges of these materials are mostly related to their morphology and mechanical properties. Enhancements in these areas are critical for applications in tissue engineering, wound healing, and drug delivery systems, which require improved mechanical properties and controlled degradation rates.

This review highlighted the main properties of PLA and its advantages in various applications, with a focus on medical uses. It also discussed the main properties of the SBS method, its processing parameters, and areas for future improvement, such as research and development in other strategic areas, such as sensors and MEMS. The continuous exploration of SBS- and PLA-based nanofibers offers significant benefits, such as reducing plastic waste and enhancing healthcare solutions. The future of this research holds the potential for substantial industrial and health impacts, driving innovation in sustainable and high-performance nanofibrous materials and advancing the SBS method.

## Figures and Tables

**Figure 1 polymers-16-03044-f001:**
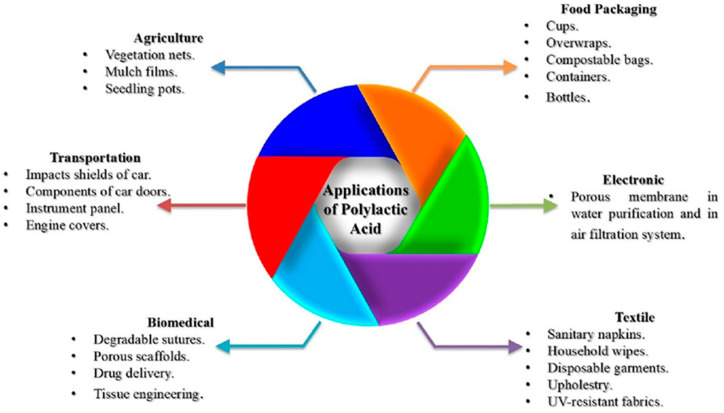
Scheme illustrating some areas of application of PLA-based materials [10].

**Figure 2 polymers-16-03044-f002:**
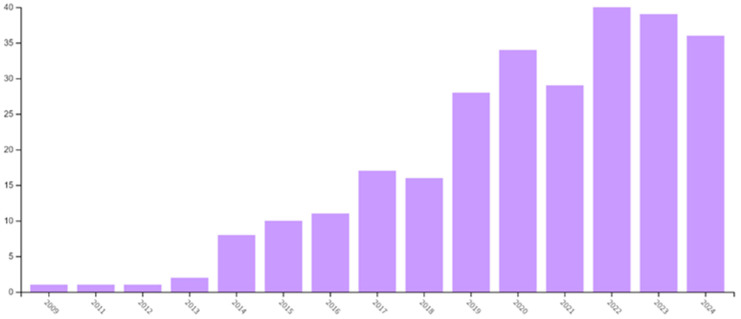
Number of scientific papers per year published on “Solution Blow Spinning” since 2009 (data from Web of Science, September 2024).

**Figure 3 polymers-16-03044-f003:**
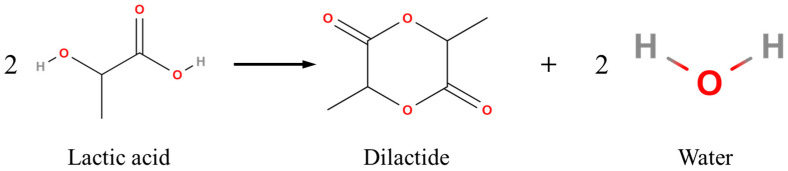
Dimerization of lactic acid.

**Figure 4 polymers-16-03044-f004:**
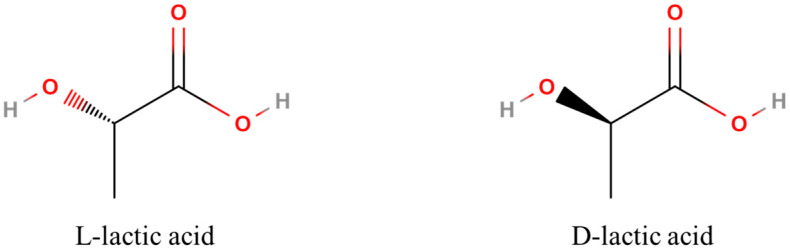
Chemical structure of L (+)- and D (−)-lactic acids.

**Figure 5 polymers-16-03044-f005:**
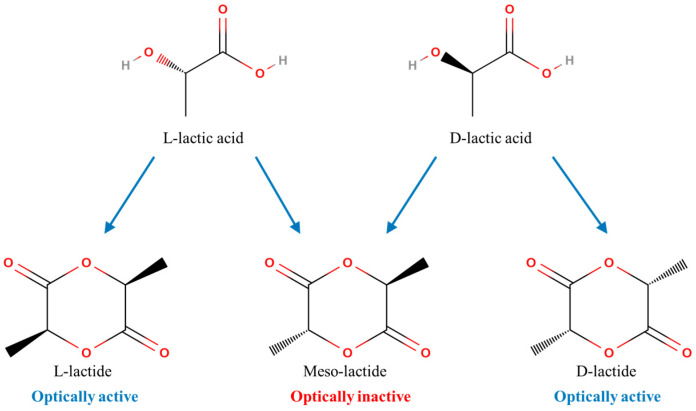
Chemical structures of LL-, meso-, and DD-lactides.

**Figure 6 polymers-16-03044-f006:**
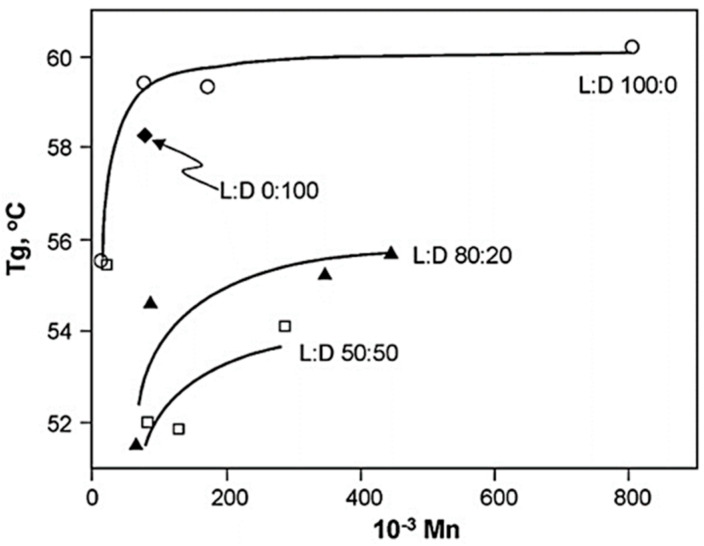
Relationship between the glass transition temperature (*T_g_*) of PLA and its L−content and molecular weight [21].

**Figure 7 polymers-16-03044-f007:**
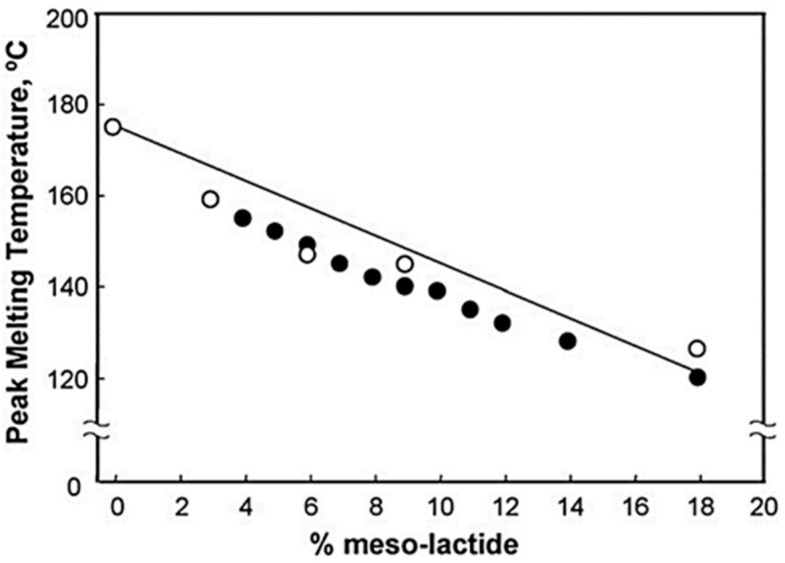
Variation of the *T_m_* as a function of % meso-lactide introduced in the PLA, (o) Represents values reported by Witzke [28]. (●) represents values reported by Hartmann [29]. The solid line is calculated based on Equation (1) (as described in [21]).

**Figure 8 polymers-16-03044-f008:**
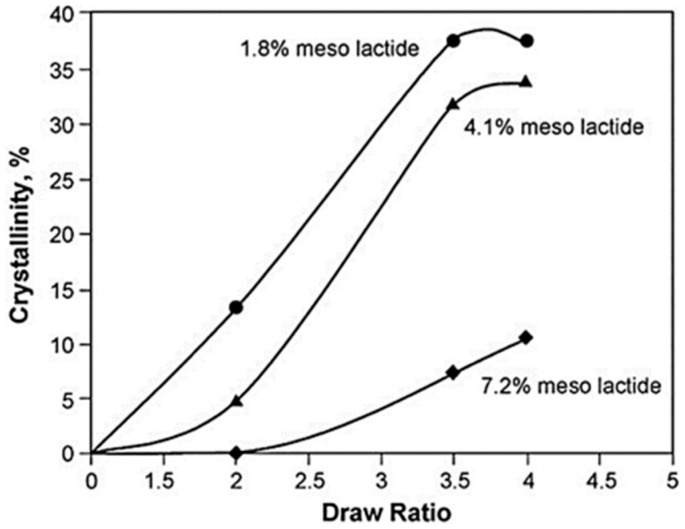
Development of crystallinity in biaxially stretched PLA at 80 °C using 100% s^−1^ strain rate [21].

**Figure 9 polymers-16-03044-f009:**
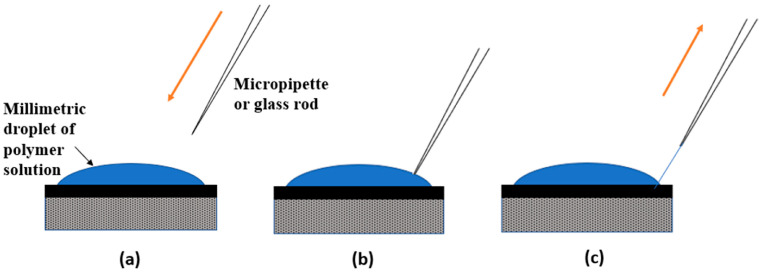
Drawing technique—production of nanofibers by contacting a previously deposited polymer solution droplet with a sharp tip and drawing it as a liquid fiber [55]: (**a**) Droplet of polymer solution; (**b**) the tip of the micropipette or glass rod is brought into contact with the droplet and (**c**) the micropipette or glass rod is withdrawn slowly away from the droplet leading to the formation of the nanofibers.

**Figure 10 polymers-16-03044-f010:**
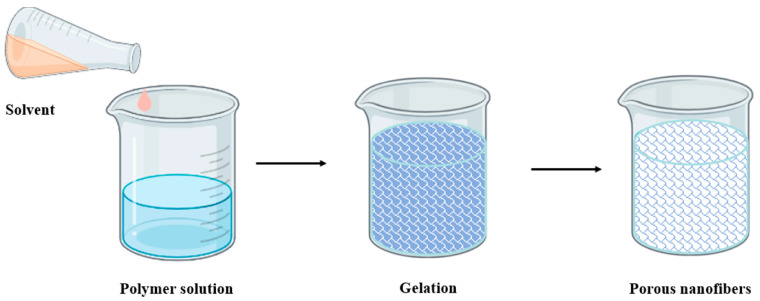
Phase separation method for producing nanofibers [55].

**Figure 11 polymers-16-03044-f011:**
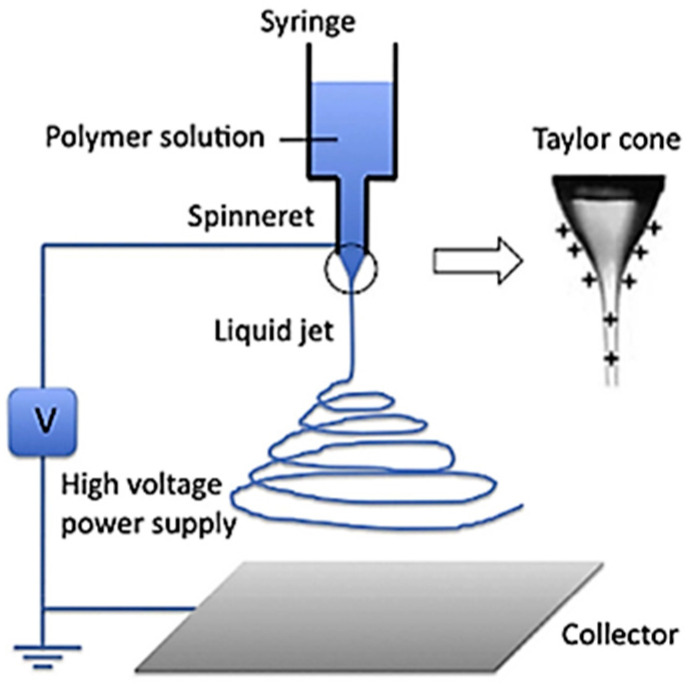
Principe of a basic electrospinning process [60].

**Figure 12 polymers-16-03044-f012:**
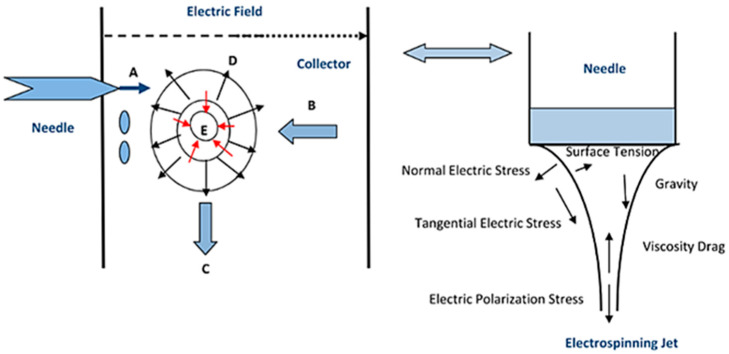
Forces affecting the charged droplet during the electrospinning process. A—electrostatic force, B—drag force, C—gravity, D—repulsion force, E—surface tension and viscoelastic force [62].

**Figure 13 polymers-16-03044-f013:**
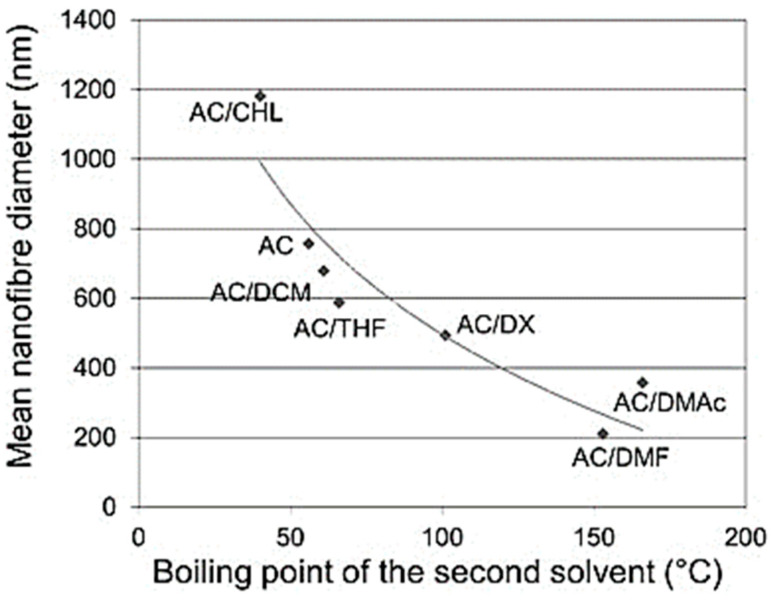
Effect of the boiling point of the second solvent in binary solvent systems on mean nanofiber diameter (acetone, AC; dichloromethane, DCM; chloroform, CHL; 1,4-dioxane, DX; tetrahydrofuran, THF; dimethylformamide, DMF; dimethylacetamide, DMAc) [80].

**Figure 14 polymers-16-03044-f014:**
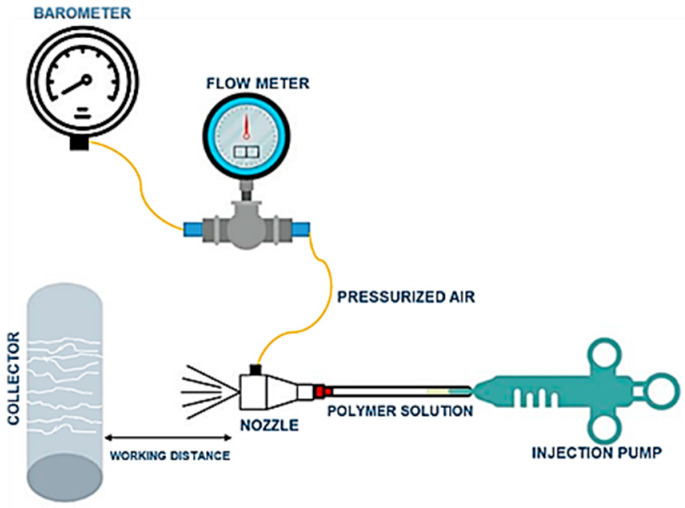
Schematic representation of nanofiber processing, illustrating the main elements of a general solution blow spinning device.

**Figure 15 polymers-16-03044-f015:**
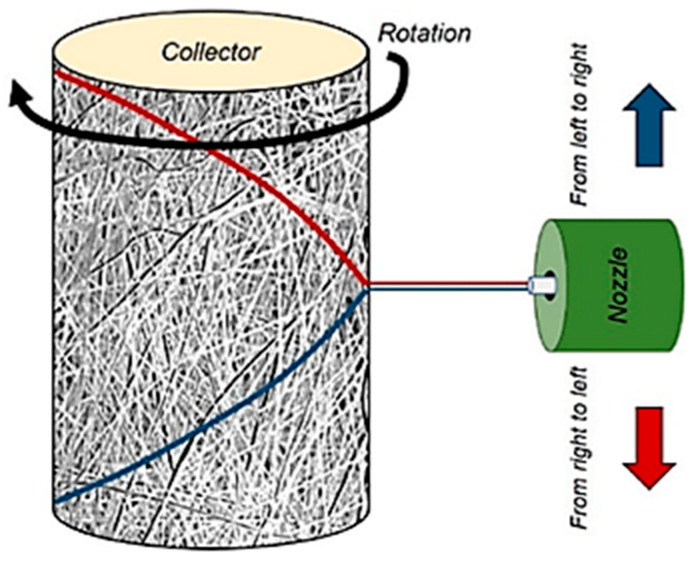
Bimodal distribution of fiber orientation in the SBS process [90].

**Table 1 polymers-16-03044-t001:** Different *T_g_* and *T_m_* of selected PLA copolymers [24].

Copolymer Ratio	*T_g_* (°C)	*T_m_* (°C)
100/0 (L/D, L) −PLA	63	178
95/5 (L/D, L) −PLA	59	164
90/10 (L/D, L) −PLA	56	150
85/15 (L/D, L) −PLA	56	140
80/20 (L/D, L) −PLA	56	125

**Table 2 polymers-16-03044-t002:** Properties of different PLA solvents significant in ES processing [66].

Solvent	Boiling Point (°C)	ε	δ Total (MPa^1/2^)	γc
Acetone	56	20.6	19.7	9.7
Chloroform	61	4.8	1.5	1.5
Ethanol	78	22.4	18.7	18.7
Dimethyl Sulphoxide	189	46.6	7.7	7.7

## Data Availability

All data are contained within the article.

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
