# Peer review of "Key Advances in Solution Blow Spinning of Polylactic-Acid-Based Materials: A Prospective Study on Uses and Future Applications"

_polymers, 2024, doi:10.3390/polym16213044_

Round 1
Reviewer 1 Report
Comments and Suggestions for Authors
1. In the review article “Key Advances in Solution-Blown Spinning of Poly (lactic acid)-Based Materials: A Prospective Study on Uses and Future Applications” the authors turn their attention on theoretical aspect of Polylactic acid (PLA)- based materials, focusing on its properties associated to structural characteristics as well as on the properties of nanofibrous materials, particularly PLA-based nanofibers, and their new applications.
The authors described various methods for producing fibrous materials, with particular emphasis on Electrospinning (ES) and Solution blow spinning (SBS) and discussed in more detail the main properties of the SBS method, along with its processing conditions and potential applications. In the manuscript the importance and relevance of the problem is clearly stated by the authors.
2. This manuscript addresses a critical issue concerning the fabrication and application of PLA-based materials. Describing the role of PLA-based materials in the field of drug delivery and wound healing, it is important to mention that their small pores and high specific area make them very interesting in the applications of wound healing and drug delivery.
2.1 To improve the manuscript, the authors should expand the Conclusions chapter. I propose to point out the specific advantages of PLA-based materials in various applications, with a focus on medical use.
2.1.1. The authors also should describe in Conclusions chapter in more detail the benefits and prospects of PLA materials in wound healing, tissue engineering, and food packaging.
2.2. In the authors' opinion, what may be the reason (reasons) for further improvement of the SBS method?
3. In the review, the authors described, summarized and compared various nanofiber fabrication techniques such as template synthesis, self-assembly, drawing, electrospinning with phase separation, and solution-blowing spinning. The authors also point out the versatility in fabricating nanofibers from a wide range of materials including polymers, metals, composites, and ceramics. I believe that the review article may prove useful in the development of PLA-based materials engineering.
4. In the review article, the authors analyzed the experimental results presented in publications cited by the authors in an extensive reference list. The authors noted that the absence of a high electric field may be an advantage of the SBS method, since it does not require specific interactions between solvents and does not require a high electric field. Another advantage of SBS over ES, for example, is the ability to achieve a higher production rate of PLA-based nanofibers and films while maintaining a quality comparable to that obtained with ES. In addition, the authors report that SBS is a scalable method that contributes to the expansion of industrial material as well as PLA-based nanofibers production, allowing for large-scale material production.
5. In the review article, the authors consider Solution blow spinning as a versatile and cost-effective technology for producing nanofibers based on a biodegradable and biocompatible polymer such as PLA. The authors note that SBS technology is based on the principles of a spinning method such as electrospinning, which produces very fine and thin fibers with controlled morphology. This review also focuses on the properties of PLA-based materials related to characteristics such as crystallinity and thermal behavior. In addition, the review focuses on the properties of nanofiber materials, particularly PLA-based nanofibers, and new applications where it is envisioned that they may be useful, such as drug delivery systems, wound healing, tissue engineering, and food packaging.
6. I believe that the references used by the authors in the paper are relevant and appropriate to the context of the review article.
7. The figures and tables presented in the manuscript are illustrative and clearly show the phenomena described by the authors. The authors indicate the sources that served as a basis for their systematization.
Author Response
Comments 1:
Answer. Thank you for the comment. We very much appreciate the positive input given by the reviewer.
Comments 2:
2.1.
Answer: Thank you for the suggestion. The following text was added to the Conclusions section to accomplish this proposal.
The versatility of PLA is especially prominent in medical applications such as tissue engineering, wound healing, and drug delivery systems, emphasizing its capacity to enhance healthcare solutions. In the contexts of wound healing and tissue engineering, PLA nanofibers can serve as scaffolds that promote cell growth and offer controlled degradation rates. Additionally, in drug delivery applications, PLA tunable degradation and release profiles significantly improve therapeutic outcomes by facilitating precise control over drug release rates.
2.1.1.
Answer: Thank you for the suggestion. To accomplish this proposal, we have expanded the conclusions section to provide a more detailed analysis of the use of PLA in food packaging applications. The following text was added to the Conclusions section.
In addition to medical uses, PLA-based materials have substantial potential in food packaging. PLA’s biocompatibility, biodegradability, and ability to form effective barrier properties make it a strong candidate for sustainable packaging solutions. Its use in food packaging can extend shelf life, reduce plastic waste, and enhance food safety through the incorporation of antimicrobial or other functional agents. Furthermore, the controlled degradation of PLA ensures that these packaging materials contribute to environmental sustainability, aligning with global efforts to minimize plastic pollution.
2.2.
Answer: Thank you for the comment.
There are several factors that may drive the need for further improvements in the Solution Blow Spinning (SBS) method. One major challenge is achieving precise control over fiber morphology and uniformity, which is essential for optimizing materials for applications such as tissue engineering, wound healing, and drug delivery. Enhancing the mechanical properties of SBS-produced nanofibers—such as tensile strength and elasticity—is also critical for expanding their use in applications that require long-term stability and functionality. Moreover, optimizing polymer-solvent systems is necessary to produce nanofibers with desired morphological, thermal, and degradation characteristics.
We believe that all these factors were considered in the last two paragraphs of the conclusions section.
Comments 3:
Answer: Thank you for your valuable feedback. We appreciate your positive remarks on the scope of our review. Taking this feedback into account and the reviewer's previous comments, we have added more detailed information in the Conclusions section, particularly focusing on the practical applications of PLA-based materials in areas such as medical use, food packaging, and sustainability. We hope that this helps to further highlight the significance of PLA in material engineering and reinforce its potential in a variety of industrial and healthcare applications.
Comments 4:
Answer: Thank you for the highlights.
Comments 5:
Answer: Thank you for your comments. We appreciate the recognition of Solution Blow Spinning (SBS) as a versatile and cost-effective technology for producing PLA-based nanofibers.
Comments 6:
Answer. Thank you for the comment. We very much appreciate the reviewer's positive input. However, according to the reviewers' comments, we have added some other references.
Comments 7:
Answer. Thank you for the comment and positive feedback.
Reviewer 2 Report
Comments and Suggestions for Authors
This review offers a comprehensive look at the Solution Blow Spinning (SBS) technique and its applications, especially in the context of producing nanofibrous materials using polylactic acid (PLA). The SBS method emerges as a versatile and cost-effective alternative to traditional spinning methods like Electrospinning (ES). PLA, a biodegradable and biocompatible polymer, finds extensive use in various fields due to its desirable properties. The review not only provides a theoretical background on PLA but also delves into different methods of producing fibrous materials, with a particular focus on SBS and ES. It details the properties and processing conditions of SBS, along with exploring the potential applications of nanofibrous materials, especially PLA-based nanofibers in areas such as drug delivery, wound healing, tissue engineering, and food packaging. Overall, this review highlights the significant potential of SBS and PLA-based nanofibers in new applications and provides valuable insights for future research directions to overcome existing challenges and improve the quality of fibrous materials. However, the paper requires improvement before publication.
1) The introduction section is weak, other methods should be briefly introduced, especially bubble electrospinning, which has certain advantages over Solution Blow Spinning. For example, it may offer better control over fiber morphology and diameter distribution. It can potentially produce more uniform nanofibers, which is beneficial for applications that require precise fiber characteristics. Moreover, bubble electrospinning might have a simpler setup in some cases, reducing complexity and cost. However, the specific advantages can vary depending on the application and processing conditions.
2) Solution Blow Spinning is a promising technique for fabricating piezoelectric nanofibers. It offers cost-effectiveness and versatility. The nanofibers produced can be integrated into MEMS devices for various applications. For instance, Piezoelectric Biosensor based on ultrasensitive MEMS system is the most advanced application of nanofibers. Additionally the nanofiber’s property will affect the periodic solution of the micro-electromechanical system.
3) There are 129 references, but lacks some important references.
4) Future research on Solution Blow Spinning and PLA-based nanofibers could focus on optimizing processing parameters for enhanced fiber quality. Investigating new applications in emerging fields and improving biocompatibility are also crucial. Additionally, exploring sustainable raw materials would be a valuable frontier.
5) English is good, but effort is still needed for further improved.
Author Response
Comments 1:
Answer: Thank you for the comment. Considering the reviewer’s comment the following text and references were added.
Bubble electrospinning is a variation of the traditional electrospinning process, designed as an alternative to producing nanofibers more efficiently [90,91]. While both methods rely on the application of a high-voltage electric field to create fibers from a polymer solution, bubble electrospinning introduces a unique mechanism to produce nanofibers [92]. The basic working principle of bubble electrospinning is the formation of polymer bubbles instead of the Taylor cone observed in electrospinning. The electrical force coming from a high-voltage power supply should be set so that it overcomes their surface tension thus causing multiple jets to eject leading to the formation of the fibers in the collector [91,92]. A key challenge in this method is controlling the number, size, geometry, and stability of the bubbles, which has led researchers to propose "critical bubble spinning," aiming to utilize a single, independent, and stable bubble for improved fiber production [93]. Bubble electrospinning can accommodate a wider range of polymer solutions, making it a versatile method for nanofiber fabrication. Its ability to produce nanofibers at a larger scale makes it particularly attractive for mass production of nanofibrillar materials for applications in fields such as biomedical engineering, filtration, and energy storage, where high volumes of nanofibers are often required [91].
Comments 2:
Answer: Thank you for the comment and thoughtful feedback highlighting the significance of Solution Blow Spinning (SBS) as a viable technique for producing piezoelectric nanofibers, emphasizing its cost-effectiveness and versatility. Indeed, the production of nanofibers to be used in micro-electromechanical systems (MEMS) is a promising research area present in a wide range of applications. One notable application mentioned is the development of ultrasensitive piezoelectric biosensors, which represent a cutting-edge use of these nanofibers in advanced technology.
In the last paragraph of the conclusions section, one sentence was added to point out this research area.
Comments 3:
Answer: Thank you for the comment. Some references were added when revising the paper.
Comments 4:
Answer: Thank you for the comment. The input is highly valuable for future research projects.
Comments 5:
Following the suggestion given by the reviewer 2 English has been revised.
Round 2
Reviewer 2 Report
Comments and Suggestions for Authors
This version is indeed suitable for publication. However, if the editor requests a revision, incorporating the following point could further engage the audience. Regarding its “Future Applications,” suggesting its application in MEMS devices is highly promising. Poly(lactic acid) is renowned for its biodegradability and biocompatibility, qualities that are extremely desirable for numerous MEMS applications where devices may come into contact with biological systems. The capacity to precisely control the spinning process can result in the creation of microscale fibers with specific properties that can be integrated into MEMS sensors or actuators. Piezoelectric nanofibers are currently widely utilized as piezoelectric biosensors in ultrasensitive MEMS systems. This indicates the potential for the solution-blown spinning of poly(lactic acid)-2 based materials to contribute to the development of advanced MEMS devices with enhanced functionality and biocompatibility.
Author Response
We fully agree with this suggestion and greatly appreciate the comment, as the electromechanical properties of PLA hold significant potential and warrant further investigation. Including a paragraph on this topic in the review is essential to encourage more researchers to explore this particularly relevant and emerging topic.
The following texts with two additional references have been added in two parts of the revised version:
Section 5. Applications of PLA nanofibrous materials
Micro-electro-mechanical systems (MEMS) – They are systems that operate at the micro or submicrometric scale, capable of exhibiting mechanical or electrical responses when subjected to electrical or mechanical perturbations, respectively [134]. As previously mentioned, polylactic acid (PLA) can demonstrate piezoelectric properties, making it particularly advantageous for integration into more complex MEMS devices utilized as sensors or actuators. Despite their ubiquitous presence in daily life, recent years have seen extensive research focused on the application of MEMS in the biomedical field [135]. This is a domain where PLA could be particularly beneficial due to its biocompatibility.
Section 6. Conclusions and Future Prospective
Additionally, coupled with the potential piezoelectric properties of PLA, these characteristics are highly desirable for numerous micro-electro-mechanical systems (MEMS) applications, particularly where devices may interface with biological systems, such as for monitoring vital signs or wearable health devices. The ability to precisely control the spinning process enables the fabrication of microscale fibers with specific structures, thus tailoring properties that can be integrated into MEMS sensors or actuators. Piezoelectric nanofibers are currently extensively used as piezoelectric biosensors in ultrasensitive MEMS systems, highlighting the potential of solution-blown spinning of poly(lactic acid)-based materials to support the development of advanced MEMS devices with enhanced functionality and biocompatibility.